# The Genome-Wide Identification of the Dihydroflavonol 4-Reductase (DFR) Gene Family and Its Expression Analysis in Different Fruit Coloring Stages of Strawberry

**DOI:** 10.3390/ijms25189911

**Published:** 2024-09-13

**Authors:** Li-Zhen Chen, Xue-Chun Tian, Yong-Qing Feng, Hui-Lan Qiao, Ai-Yuan Wu, Xin Li, Ying-Jun Hou, Zong-Huan Ma

**Affiliations:** 1State Key Laboratory of Aridland Crop Science, Gansu Agricultural University, Lanzhou 730070, China; 19119301698@163.com; 2College of Horticulture, Gansu Agricultural University, Lanzhou 730070, China; 13383266259@163.com (X.-C.T.); 18893425315@163.com (Y.-Q.F.); 13893459314@163.com (H.-L.Q.); 18893011423@163.com (A.-Y.W.); lixin@gsau.edu.cn (X.L.); houyj@gsau.edu.cn (Y.-J.H.)

**Keywords:** forest strawberry, dihydroflavonol 4-reductase (DFR), anthocyanin, expression analysis

## Abstract

Dihydroflavonol 4-reductase (DFR) significantly influences the modification of flower color. To explore the role of *DFR* in the synthesis of strawberry anthocyanins, in this study, we downloaded the CDS sequences of the *DFR* gene family from the *Arabidopsis* genome database TAIR; the DFR family of forest strawberry was compared; then, a functional domain screen was performed using NCBI; the selected strawberry DFR genes were analyzed; and the expression characteristics of the family members were studied by qRT-PCR. The results showed that there are 57 members of the DFR gene family in strawberry, which are mainly expressed in the cytoplasm and chloroplast; most of them are hydrophilic proteins; and the secondary structure of the protein is mainly composed of α-helices and random coils. The analysis revealed that *FvDFR* genes mostly contain light, hormone, abiotic stress, and meristem response elements. From the results of the qRT-PCR analysis, the relative expression of each member of the *FvDFR* gene was significantly different, which was expressed throughout the process of fruit coloring. Most genes had the highest expression levels in the full coloring stage (S4). The expression of *FvDFR30*, *FvDFR54*, and *FvDFR56* during the S4 period was 8, 2.4, and 2.4 times higher than during the S1 period, indicating that the *DFR* gene plays a key role in regulating the fruit coloration of strawberry. In the strawberry genome, 57 members of the strawberry *DFR* gene family were identified. The higher the *DFR* gene expression, the higher the anthocyanin content, and the *DFR* gene may be the key gene in anthocyanin synthesis. Collectively, the *DFR* gene is closely related to fruit coloring, which lays a foundation for further exploring the function of the *DFR* gene family.

## 1. Introduction

*Fragaria vesca*, commonly known as the woodland strawberry, stands out not only for its exquisite taste but also for its remarkable nutritional profile, boasting a rich array of vitamins. Revered as the “queen of fruit”, it has been traditionally acknowledged for its digestive, heat-alleviating, and thirst-quenching properties. Beyond its culinary appeal, this fruit also has powerful health benefits. Anthocyanins, abundant in *Fragaria vesca*, contribute to its vibrant color and exhibit robust antioxidant properties, playing a pivotal role in preventing and treating various chronic diseases [1]. *DFR* is a key enzyme in the anthocyanin biosynthesis pathway, which enables dihydroflavonol to produce colorless anthocyanins [2], and was initially isolated in maize and goldfish by Reilly et al. in 1985 [3]. Positioned as the foremost enzyme in the anthocyanins biosynthetic pathway, *DFR* significantly influences the modification of flower color [4]. *DFR* exhibits catalytic prowess in transforming colorless dihydroflavonol substrates such as dihydroquercetin (DHQ), dihydroacricetin (DHM), and dihydrokaempferol (DHK) into distinct pigments, contributing to the diversity of colors observed in plants like *Centaurea cyanus* Linn. (Cornflower), *Consolida ajacis* (L.) *Schur* (Delphinidin), and *Pelargonium hortorum Bailey* (Pelargonium) [5,6,7]. These colored anthocyanins serve as precursors to anthocyanins, and their content plays a role in influencing the production of anthocyanins to some extent [8].

The involvement of DFR is not limited to anthocyanin production and is intricately linked to the synthesis of flavonoids. Flavonoids are mainly found in plants in the form of glycosides, which can not only make plants appear in different colors such as blue, red, pink, etc., but also have the functions of protecting plants from ultraviolet rays, resisting biotic and abiotic stresses, and regulating the secretion and transport of plant hormones [9]. The multifaceted role of flavonoids encompasses vital functions in plant growth, development, disease resistance, stress resilience, and the formation of high-quality produce [10,11]. Furthermore, the health benefits associated with flavonoids extend beyond their presence in plants, demonstrating promising potential in suppressing tumors and preventing cardiovascular diseases [12].

Notably, studies on the *LaDFR1* gene in *Lycoris albiflora* have revealed an increase in expression with the extension of flowering time. This pattern aligns with changes in flower color phenotype and the overall trend of total anthocyanin content during flowering [13]. Qin et al. found that *HvDFR* was constitutively expressed in all tissues and expressed highly in flower as well as was positively correlated with anthocyanin content [14]. The expression of *MaDFR1*, *MaDFR2a*, and *MaDFR2b* in *Muscari* spp. was higher, and these findings suggest that the *DFR* gene expression pattern is consistent with the coloration of floral organs [15].

It is essential to note that the expression of the *DFR* gene and its intensity are pivotal factors in anthocyanin synthesis in fruit. At different stages of fruit development, the expression level of the *DFR* gene is higher in black currants than that in red and white currants [16]. As the fruit diameter increases and the skin color deepens, the expression of the *DFR* gene in black currants shows an upward trend. In red strawberries, the gene expression level of the *DFR* regulatory gene is significantly higher than that in white strawberries [17]. In mango studies, the expression of the *DFR* gene was different in different colored mango peels, with higher expression in green peels and lower expression in yellow peels [18]. This suggests that the *DFR* gene may be involved in the regulation of anthocyanin synthesis, thus affecting fruit color.

Although the function of the *DFR* gene in regulating color formation has been studied in many plants, the identification of *DFR* gene family members and their expression characteristics in different coloration stages of forest strawberry have not been reported. The expression of the *DFR* gene family in strawberry was studied by real-time fluorescence quantitative PCR, and the key candidate family members involved in the regulation of strawberry anthocyanins were identified, which provided a basis for subsequent functional identification.

## 2. Results

### 2.1. Sequence Characteristics of the Strawberry DFR Gene Family

A homologous alignment search of strawberry databases obtained 57 strawberries from the *DFR* gene family, which were named *FvDFR1–FvDFR57*. According to the physicochemical properties of the strawberry DFR proteins (Table 1), the number of amino acids is between 69 and 1253, where the FvDFR16 amino acid sequence is the shortest and FvDFR49 the longest. The molecular weight is 7686.98–136,295.17 Da. The theoretical isoelectric point is 4.69–9.43. The average hydrophilicity is between −0.231 and 0.249. The instability coefficient *FvDFR36* has the smallest value of 14.46, and *FvDFR56* has a value of 46.13.

TBtools (v.1.113) software was used for chromosome localization analysis. In total, 57 *FvDFR* gene members were distributed on six chromosomes of strawberry: 6 genes were distributed on chromosome 2, 7 genes were distributed on chromosome 3, 3 genes were distributed on chromosome 4, 13 genes were distributed on chromosome 5, and 4 genes were distributed on chromosome 6. There are 24 genes on chromosome 7 and no genes on chromosome 1. Among them, the gene members distributed on chromosome 7 were the most, accounting for 42.1%, and the gene members distributed on chromosome 4 were the least, accounting for 5.2% (Figure 1).

### 2.2. Subcellular Localization and Secondary Structure Analysis of the Strawberry DFR Gene Family

The subcellular localization expression of the strawberry *DFR* gene family was mainly concentrated in the cytoplasm and chloroplasts (Table 2). Secondly, it is expressed in the nucleus, cell membrane, and mitochondria. It is slightly expressed in the peroxisome, with only eight genes located in the peroxisome. The expression of each gene in different organelles is related to the structure and function of this gene. A secondary structure analysis of the strawberry *DFR* gene family (Table 3) showed that the gene family was mainly composed of alpha helices (23.19–51.59%) and random coils (23.89–44.29%) in the 57 coding proteins, followed by extended strands (11.8–31.88%) and less beta turns (3.27–11.01%).

### 2.3. Analysis of Gene Structure, Motif, and cis-Acting Elements of Strawberry DFR Gene Family

MEME online (http://meme-suite.org/tools/meme, accessed on 4 June 2024) software was used to predict the strawberry family protein motifs for 10 motifs (Table 4). *FvDFR16* contains only motif3, and *FvDFR51* contains only two motifs, motif4 and motif8. The N terminus of most sequences is motif3, and the C termini are all motif10, of which *FvDFR36* and *FvDFR45* share the same motif and contain only motif1, motif3, and motif9. A total of 18 genes in *FvDFR1*, *FvDFR3*, *FvDFR8*, *FvDFR12*, *FvDFR13*, *FvDFR14*, *FvDFR24*, *FvDFR26*, *FvDFR27*, *FvDFR30*, *FvDFR37*, *FvDFR41*, *FvDFR43*, *FvDFR47*, *FvDFR48*, *FvDFR50*, *FvDFR53*, and *FvDFR55* had the same motif (Figure 2A).

The *FvDFR* gene mainly contained light, hormone, abiotic stress, and meristem response elements, and the hormone response elements contained auxin, gibberelin, abscisic acid, salicylic acid, and methyl jasmonate response elements. The abiotic stress response elements contained low temperature, drought, and anaerobic induction. After analysis, we found that all except *FvDFR21*, *FvDFR22*, and *FvDFR23* contained light response. All contained abscisic acid response elements except for *FvDFR13*, *FvDFR24*, *FvDFR29*, *FvDFR32*, and *FvDFR45*. And, the gibberellin response element appeared in the six genes *FvDFR10*, *FvDFR12*, *FvDFR13*, *FvDFR29*, *FvDFR46*, and *FvDFR55* (Figure 2C).

According to the structure of the 57 genes in the strawberry *DFR* family (Figure 2D), the number of exons is between 2 and 11, and most genes contain 6 exons. The maximum number of *FvDFR49* exons was 11, and the minimum number of exons in *FvDFR29*, *FvDFR45*, and *FvDFR51* was 2. *FvDFR16*, *FvDFR17*, *FvDFR28*, *FvDFR29*, *FvDFR34*, *FvDFR44*, *FvDFR45*, *FvDFR48*, *FvDFR49*, *FvDFR51*, *FvDFR52*, and *FvDFR53* had no upstream and downstream gene sequences. Other genes were structurally intact, and the gene structure of genes close to kinship has the same distribution and length.

### 2.4. Phylogenetic Tree Construction of the Strawberry DFR Family

According to this study, the strawberry *DFR* gene family is divided into six subfamilies according to a phylogenetic analysis (Figure 3). To further understand the evolutionary relationship between *FvDFR* and other species, this study performed a phylogenetic analysis of 7 species and 57 protein sequences in *Arabidopsis thaliana*, tomato, potato, grape, wheat, wild tobacco, and walnut. Subfamilies A, B, C, and D contain only strawberry *DFR* genes. *FvDFR2* and *FvDFR51* are separate subgroups A and B. The F subfamily contains only the *NaDFR2* gene. There are 7 species genes of *Arabidopsis thaliana*, tomato, potato, grape, wheat, wild tobacco, and walnut in subfamily E, indicating that the *DFR* family gene of strawberry in subfamily E is closely related to other species.

### 2.5. Collinearity Analysis of the Strawberry DFR Family

Through the analysis of the commonness among the *DFR* gene families of *Arabidopsis* apple, grape, and rice (Figure 4), it was found that strawberry and apple had 28 collinear gene pairs, which constituted the most pairs. It was followed by *Arabidopsis* and grape, with 14 and 17 gene pairs, respectively. There were the least six pairs of collinear genes in rice. The results showed that there were more homologous genes in strawberry and dicotyledon than in monocotyledon. A total of four collinearity relationships were found within the *DFR* gene family species (Figure 5). Located on chromosomes 3, 5, and 7, they are *FvDFR19/FvDFR57*, *FvDFR18/FvDFR56*, *FvDFR9/FvDFR8*, and *FvDFR9/FvDFR7*, respectively. The results suggest that some *FvDFR* genes may be produced by gene replication, and these genes may have similar functions.

### 2.6. Codon Preference Analysis of the Strawberry DFR Family

By calculating the RSCU value of 57 CDS sequences in the strawberry *DFR* family, there were 27 high-frequency codons (RSCU > 1) in the strawberry sucrose phosphate synthetase CDS sequence, including two codons with RSCU = 1 and one codon with RSCU > 2 (Figure 6), indicating that there is a highly preferred codon in the strawberry *DFR* gene. The AGA codon used the highest frequency, at 2.4712, and the translated alanine GCG was the lowest, at 0.3464 (Figure 7).

### 2.7. Analysis of the Expression Patterns of the DFR Gene Family in Strawberry

The expression levels of the *DFR* family in different plant organs were significantly different. *FvDFR1* is almost not expressed in many tissues and is only expressed in late anther development. The expression levels of *FvDFR10*, *FvDFR11,* and *FvDFR12* were higher in the late stage of seed and anther development. The expression level of *FvDFR32* was higher in mature seeds. The expression levels of *FvDFR30*, *FvDFR31,* and *FvDFR55* were higher in flesh and peel during ripening, but lower in other periods. The contents of *FvDFR3*, *FvDFR4*, *FvDFR5*, and *FvDFR6* in pericarp and seeds were higher in the mid-ripening stage. These genes may be involved in pigment synthesis (Figure 8).

### 2.8. Quantitative Fluorescence Analysis of Strawberry DFR Gene Family and Determination of Anthocyanin Content in Strawberry Fruit

From S1 to S4, there are different coloring stages of strawberries. According to the determination of anthocyanin content (Figure 9), the content of anthocyanin kept increasing and the content of S4 period reached 40 times that of S1 period. With the increase in fruit coloring, the content of anthocyanin gradually increased. Thus, the higher the anthocyanin content, the darker the strawberry color.

The qRT-PCR results (Figure 10) verified the transient expression levels of strawberry fruit in different periods and analyzed the significant relative expression differences between treatments. The analysis found that most of the members had the highest gene expression in the S4 period, and the expression of *DFR* genes in early fruit color stages was mostly lower than in the middle and later color stages. Among them, the expression levels of five genes, *FvDFR5*, *FvDFR30*, *FvDFR42*, *FvDFR44*, and *FvDFR55*, showed an increasing trend in general, which was consistent with the trend of anthocyanin content. The expression levels in S4 were 2, 8, 3, 7, and 4 times higher than those in S1. *FvDFR13, FvDFR16, FvDFR18*, *FvDFR24*, *FvDFR25*, *FvDFR46*, *FvDFR48*, and *FvDFR49* showed the highest gene expression in the S1 period. However, there was an upward trend from S2 to S4. These results suggest that this gene promotes anthocyanin accumulation in the S2, S3, and S4 phases and may be involved in anthocyanin regulation. In addition, *FvDFR1*, *FvDFR3, FvDFR21*, *FvDFR27*, *FvDFR32*, and *FvDFR51* showed the highest gene expression levels in the S3 period. Overall, the expression level increased first and then decreased. The expression levels of these six genes in S3 were 1.3, 34, 2, 7, 7 and 3.5 times of those in S1, respectively. This suggests that the gene is highly expressed during the rapid accumulation of fruit pigment and may regulate anthocyanin synthesis.

### 2.9. Protein Interaction Network Analysis and Prediction of Strawberry DFR Gene Family

The interactions between the 57 FvDFR proteins were predicted by the STRING website. The results display, FvDFR1, FvDFR2, FvDFR6, FvDFR8, FvDFR9, FvDFR12, FvDFR13, FvDFR16, FvDFR23, FvDFR24, FvDFR26, FvDFR27, FvDFR28, FvDFR29, FvDFR31, FvDFR32, FvDFR33, FvDFR48, FvDFR50, FvDFR54, FvDFR55, and FvDFR57; these 22 FvDFR proteins were identified by 4-hydroxycoumarin synthase (XP_004292734.1), trans-cinnamate 4-monooxygenase (XP_004294725.1), naringenin (XP_004287814.1), leucoanthocyanidin dioxygenase (XP_004298720.1), and 4-coumarate-CoA (XP_004310219.1) ligase interact to form a protein interaction network (PPI network) (Figure 11).

## 3. Discussion

At present, there is still a gap in the research on the *DFR* gene in strawberry at home and abroad. In this study, qRT-PCR was used to study the expression characteristics of this family member. The results showed that there were 57 members of the *DFR* gene family in strawberry, which were mainly expressed in the cytoplasm and chloroplast; most of them were hydrophilic proteins; and the secondary structure of the protein was mainly composed of α-helices and random coils. The analysis revealed that *FvDFR* genes mostly contain light, hormone, abiotic stress, and meristem response elements. From the results of the qRT-PCR analysis, the relative expression of each member of the *FvDFR* gene was significantly different, which was expressed throughout the process of fruit coloring. Most genes had the highest expression levels in the full coloring stage (S4).

Dihydroflavonol-4-reductase (DFR) catalyzes the first reaction leading to anthocyanins, and it is considered a key regulatory enzyme of anthocyanin biosynthesis in plants [9]. The *DFR* gene family has been identified in multiple species, such as the Brassica rapa [19], grapes [20], turnip [21], turnip radish [22], and lily [23]. The DFR gene family may also be present in other plants. In this study, 57 members of the *DFR* gene family were identified in a strawberry genome database using the bioinformatics method. However, compared with the 26 members of rape [24] and the 96 members of apple [25], the number of *DFR* gene families in strawberry is larger, reflecting the large differences between different species. The physicochemical properties, subcellular localization, and *cis*-acting elements of the *DFR* gene family were then predicted and analyzed. According to the protein sequence analysis, 68.4% of *DFR* family members are hydrophilic proteins and 85.9% are stable proteins. In their secondary structure, α-helices and random coils are the main secondary structure elements; when compared with the study by Hu Yong [26], the study findings were consistent. This experimental subcellular localization analysis revealed that it was mainly expressed in the cytoplasm. *DFR* gene expression varies in different species, such as soybean GmDFR protein being mainly found in the nucleus (34.8%) and the cytoplasm (21.7%) [27], perhaps because the distant relationship between species leads to different results. But, the *Paeonia delavayi DFR* protein is mainly found in the cytoplasm [28], and the *Ophiorrhiza japonica DFR3* protein is most likely to locate in the cytoplasm [29]. These conclusions are all consistent with the conclusions of this paper. A fhylogenetic analysis of the strawberry *DFR* family divided these 57 genes into six subfamilies, and other species were only found in subfamilies E and F, indicating close proximity to them. In general, genes located in the same clade can be speculated to have similar biological functions if they have sequences with high similarity to each other.

From the expression of the *DFR* gene family in various tissues, we showed that *FvDFR9*, *FvDFR10*, *FvDFR11*, *FvDFR12*, *FvDFR14*, and *FvDFR15* were all high in the anther period. *FvDFR3*, *FvDFR4*, *FvDFR5*, *FvDFR6*, *FvDFR7*, and *FvDFR8* were high in the pericarp period. It is found that *DFR* genes in different species are differently expressed in different tissues at different periods. For example, the *AcDFR* gene in aconite is mainly expressed in flowers, and it is hardly expressed in roots, stems, and leaves [29]; with the initial *DFR* gene in the calyx to the full bud period, the relative expression of *DFR* is increasing, especially from the initial flowering period to the full flowering period [30], consistent with the findings presented here.

Dihydroflavonol-4-reductase (DFR) is a key enzyme in the third phase of anthocyanin synthesis that catalyzes the formation of colorless anthocyanin by dihydroflavonols [31]. According to the qRT-PCR results, most genes had the highest expression during the S4 period. In the determination of anthocyanin content, the anthocyanin content was the highest in the S4 period and the deepest fruit color in this period. In combination with the tissue expression, *FvDFR5*, *FvDFR7*, *FvDFR17*, *FvDFR44*, and *FvDFR45* all showed the highest expression in the fruit, and the higher the *DFR* gene expression, the higher the anthocyanin content. Different species and content of anthocyanin are found in different tissues of the *DFR* gene, with the higher content of anthocyanin in darker parts [32]. Based on the analysis of anthocyanin content in blueberry, the DFR gene was responsible for catalyzing the production of anthocyanins, which played a significant positive role in regulating anthocyanin synthesis. The higher the gene expression level, the higher the total anthocyanin content [33]. These results are consistent with the results of this study. It is speculated that the *DFR* gene may be the key gene in anthocyanin synthesis.

An analysis of the results showed that the seven genes, *FvDFR1*, *FvDFR3*, *FvDFR21*, *FvDFR27*, *FvDFR32*, *FvDFR48*, and *FvDFR51*, showed the highest expression in the S3 period, and the anthocyanin content showed a trend of the first increasing and then decreasing. In eggplant fruit, the relative content of anthocyanins first increased and then decreased, and the *DFR* expression of purple eggplant was consistent with the change in the relative content of anthocyanins [34], consistent with the research results of this paper. These seven *DFR* genes are highly expressed during the rapid accumulation of fruit pigment. The lowest expression of the eight genes, *FvDFR29*, *FvDFR32*, *FvDFR39*, *FvDFR45*, *FvDFR51*, *FvDFR52*, *FvDFR54*, and *FvDFR56*, was found in the fruit turn period. In safflower strawberry, the expression of four structural genes, *FpDFR1*, *FpDFR2*, *FpDFR3* and *FpANS1*, in L (bud stage), Z (color turning stage), and D (big bud stage) was the highest in the lowest bud period, but the lowest in the color turning period [35], consistent with the research results of this paper. We preliminarily showed that the *DFR* gene plays a key role in regulating the fruit coloring of strawberry.

## 4. Materials and Methods

### 4.1. Plant Materials

“Monterey” strawberry fruit was selected as the research material, and fruits from the green fruit stage (S1), 20% coloring stage (S2), 50% coloring stage (S3), and fully colored stage (S4) were collected. A total of 15 fruit samples were collected at different coloring stages, and every fifth fruit was a duplicated, accurately weighed, quickly frozen with liquid nitrogen, and stored at −80◦C for subsequent experiments.

### 4.2. Identification of the Strawberry DFR Gene Family

The CDS sequence of the *DFR* gene family was downloaded from the *Arabidopsis* genome database TAIR (https://www.arabidopsis.org/, accessed on 14 May 2024), and the forest strawberry annotation group file was downloaded using phytozome13 (https://phytozome-next.jgi.doe.gov/, accessed on 16 May 2024) [36]. The *DFR* family of forest strawberry was compared with the downloaded *Arabidopsis thaliana* using TBtools software, and 64 candidate genes were selected. Then, NCBI (https://www.ncbi.nlm.nih.gov/cdd/, accessed on 17 May 2024) was used for functional domain screening, and the candidate genes that did not contain the specific domain of the *DFR* gene (FR_SDR_e, NADP) were excluded. A total of 57 members of the strawberry *DFR* gene family were obtained after screening. The ExPASy (https://web.expasy.org/protparam/, accessed on 21 May 2024) [37] online website ProtParam tool was used to analyze the molecular weight, amino acid number, isoelectric point. and fat coefficient of the strawberry *DFR* gene after selection.

### 4.3. Chromosome Localization, Subcellular Localization, and Secondary Structure Analysis of the Strawberry DFR Gene Family

Gene Location Visualize from GTF/GFF in TBtools software was used to analyze and visualize the strawberry genome data. PRABI (https://npsa-prabi.ibcp.fr/cgi-bin/npsa_automat.pl?page=npsa_sopma.html, accessed on 25 May 2024) was used to analyze the secondary structure of the *DFR* gene family proteins, and WoLF PSORT (https://wolfpsort.hgc.jp/, accessed on 27 May 2024) was used to predict the subcellular localization of the strawberry *DFR* gene family [38].

### 4.4. Analysis of Gene Structure, Motif, and cis-Acting Elements of Strawberry DFR Gene Family

A promoter *cis*-acting element analysis was performed using PlantCARE (http://bioinformatics.psb.ugent.be/webtools/plantcare/html/, accessed on 30 May 2024) [39]. MEME (http://meme-suite.org/tools/meme, accessed on 4 June 2024) was used for a protein motif analysis. Gene Structure View (Advanced) in TBtools software was used for a gene structure analysis.

### 4.5. Phylogenetic Tree Construction and Collinearity Analysis of the Strawberry DFR Family

Annotated group files for *Arabidopsis thaliana*, grape, rice, and apple were downloaded via phytozome13 (https://phytozome-next.jgi.doe.gov/, accessed on 10 June 2024). The annotated genomic information of *Arabidopsis thaliana*, grape, rice, and apple was imported using TBtools for a comparative analysis among species. The annotation information of the strawberry genome was compared within species using MCScanX in TBtools [40]. The adjacency method (NJ) was used to construct a phylogenetic tree through One Step Build a ML Tree in TBtools, and the iTOL tool (https://itol.embl.de/, accessed on 14 June 2024) was used to beautify the tree.

### 4.6. Protein Interaction Network Analysis and Codon Preference Analysis of Strawberry DFR Gene Family

The strawberry *DFR* family protein sequence was used to construct the protein interaction network through the STRING database (https://cn.string-db.org/, accessed on 18 June 2024) [41]. The codon preference of the *FvDFR* gene was analyzed using Genepioneer (http://112.86.217.82:9919/#/, accessed on 20 June 2024). The RSCU file was then downloaded to process the data and draw the relative synonymous codon usage (RSCU) bar chart. TBtools software was used to visualize the RSCU heat map.

### 4.7. Tissue-Specific Expression Profile of the Strawberry DFR Gene Family

The tissue-specific expression profile was analyzed using Strawberry EFP Browser in BAR (https://bar.utoronto.ca/, accessed on 23 June 2024). The Primary Gene ID of strawberry was used for data analysis, and TBtools software was used for visualization.

### 4.8. Determination of Anthocyanin Content in Strawberry Fruits in Different Time Periods

A total of 1.0 g of strawberry fruit was ground in liquid nitrogen in a 10 mL centrifuge tube, washed with 1% HCl methanol solution, transferred to a test tube for scaling, mixed well, and extracted at 4 °C away from light for 20 min. It was then shaken several times throughout the period. With a 0.2 µm polyethersulfone filter membrane (Krackeler Scientific, Inc., Albany, NY, USA), the solution was filtered and the filtrate was collected. The TU-1900 double beam UV-visible spectrophotometer (Beijing Purkinje General Instrument Co., Ltd., Beijing, China) was then used to determine the absorbance value at 530 nm and 600 nm three times. The anthocyanin content (U) is represented as the difference between absorbance values at 530 nm and 600 nm per gram of fresh pericarp peel tissue, namely U = (OD_530_ − OD_600_)/gFW.

### 4.9. Quantitative RT-PCR (qRT-PCR) Analysis

The total RNA from strawberry fruit was extracted by CTAB and cDNA synthesized using the Prime Script RT reagent Kit (TaKaRa). Using the website of Biological Bioengineering (Shanghai) Co., Ltd. (https://www.sangon.com/new Primer Design, accessed on 25 June 2024), the primer was designed and synthesized for 33 strawberry *DFR* genes (Table 5). With the Light Cycler^®^ 96 Real-time PCR System (Roche, Basel, Switzerland), real-time quantitative PCR was performed with strawberry *GAPDH* as the reference gene. The amplification reaction system totalled 20 μL: ddH_2_O 7 μL, cDNA 1 μL, 1 μL of each of the upstream and downstream primers, and SYBR Premix Ex Taq 10 μL. The reaction procedure was 95 °C predenaturing for 30 s, 95 °C denaturing for 10 s, 60 °C annealing for 30 s, and 72 °C extension for 30 s, with a total of 50 cycles. The test was repeated three times. Relative gene expression was calculated using the 2^−ΔΔCt^ method [42,43] and plotted using SPSS 26 and Origin 2021.

## 5. Conclusions

In this study, 57 members of the strawberry *DFR* gene family were identified on six different chromosomes. The analysis found that most of the *DFR* gene family were hydrophilic proteins, and the protein secondary structure was dominated by α-helices and random coils. The results of qRT-PCR showed that the *FvDFR3*, *FvDFR27*, *FvDFR32*, and *FvDFR51* expression levels were the highest in the rapid fruit coloring stage. The expression levels of *FvDFR30*, *FvDFR54*, and *FvDFR56* were the highest in the complete fruit coloring stage. These genes can be used as candidate genes for further functional studies and are of great significance for further studies on the function of strawberry *DFR* family members.

## Figures and Tables

**Figure 1 ijms-25-09911-f001:**
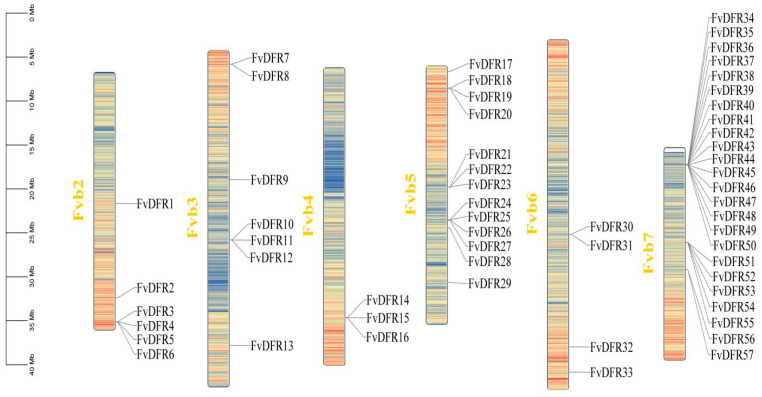
Chromosome distribution of the *DFR* genes in strawberry. The left scale indicates the chromosome length (Mb), with *DFR* gene markers on the right side of each chromosome. Different chromosomal colors indicate different gene densities, with red indicating the highest density and blue the lowest density.

**Figure 2 ijms-25-09911-f002:**
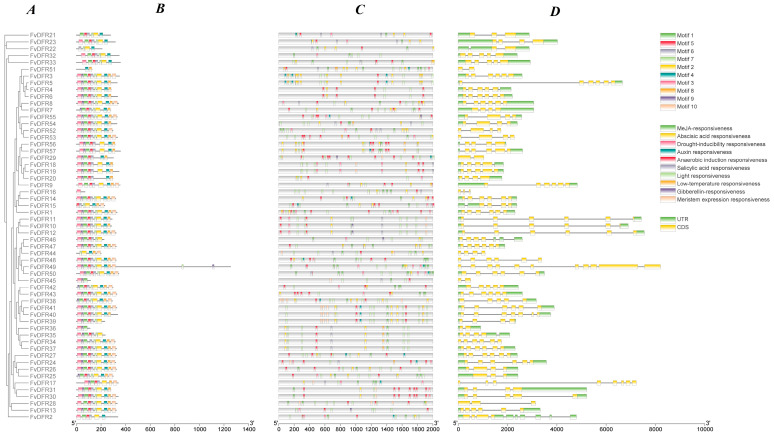
Analysis of the gene structure, motif, and *cis*-acting elements of the strawberry *DFR* gene family. (**A**,**B**) Analysis of the conserved motif of the *DFR* gene family in strawberry. (**C**) The *cis*-acting element analysis was performed for the first 2000 bp of the promoters of strawberry *DFR* gene family members. (**D**) The exon–intron structure of the *FvDFR* genes.

**Figure 3 ijms-25-09911-f003:**
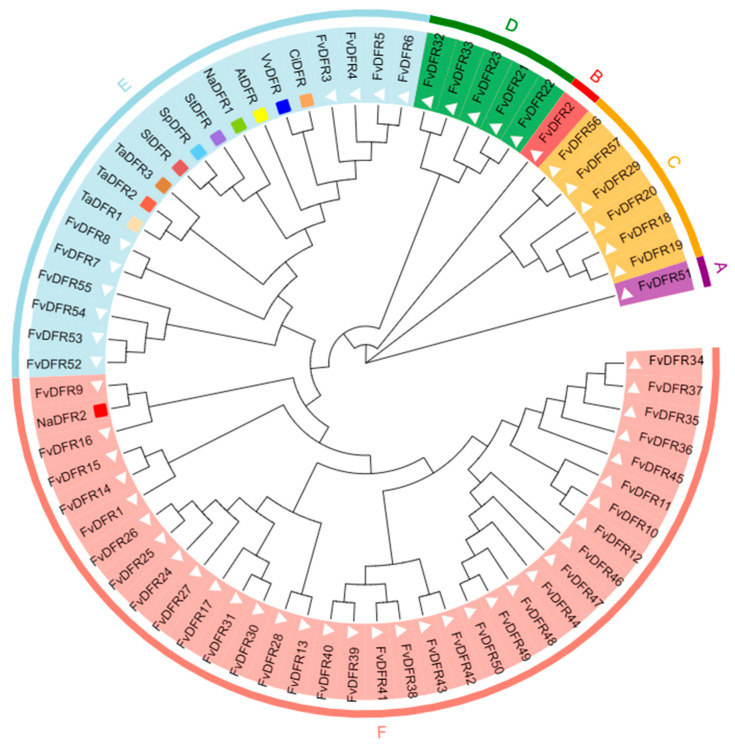
The evolutionary analysis of the *DFR* gene families. The phylogenetic tree was constructed using the *DFR* protein sequence and using the NJ method. A white triangle represents strawberry, a yellow-green rectangle represents *Nicotiana gossei* 1, a red rectangle represents *Nicotiana gossei* 2, a yellow rectangle represents *Arabidopsis thaliana*, an Indian red rectangle represents tomato, a medium purple rectangle represents potato, a blue rectangle represents grape, a light sky blue rectangle represents wild tomato, a sandy brown rectangle represents walnut, a Navajo white rectangle represents wheat 1, a tomato red rectangle represent wheat 2, and a Peru brown rectangle represents wheat 3. The phylogenetic tree is named using A, B, C, D, E, and F, and these different names represent different subfamilies.

**Figure 4 ijms-25-09911-f004:**
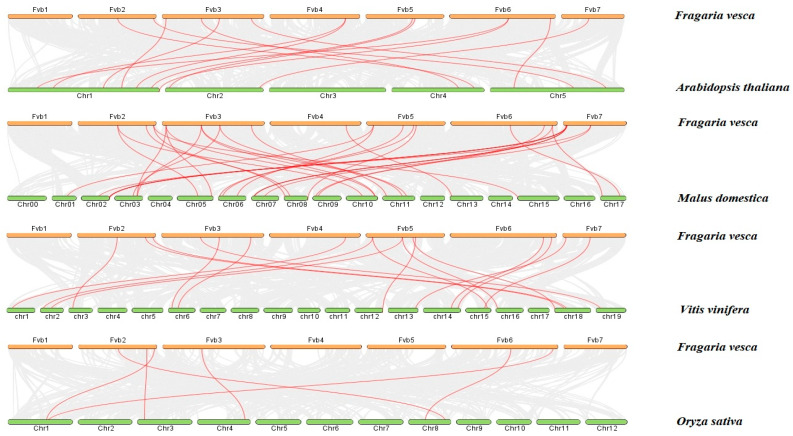
Collinearity analysis of the *DFR* gene family in *Arabidopsis*, apple, grape, and rice. The gray lines in the background represent the collinearity of the *Arabidopsis*, apple, grape, and rice genomes, while the red lines represent the gene pairs of the strawberry *DFR* genes.

**Figure 5 ijms-25-09911-f005:**
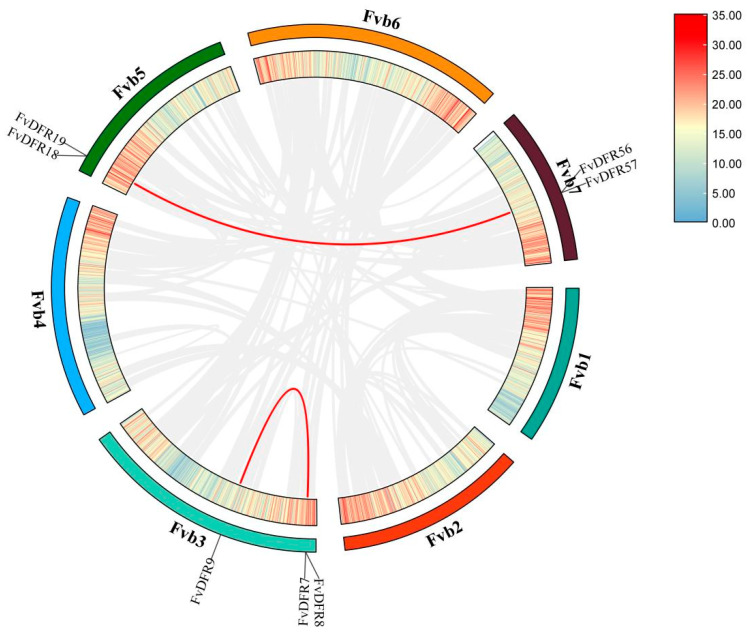
Collinearity analysis of the strawberry *DFR* gene family. Red lines represent the duplicated *FvDFR* gene pairs.

**Figure 6 ijms-25-09911-f006:**
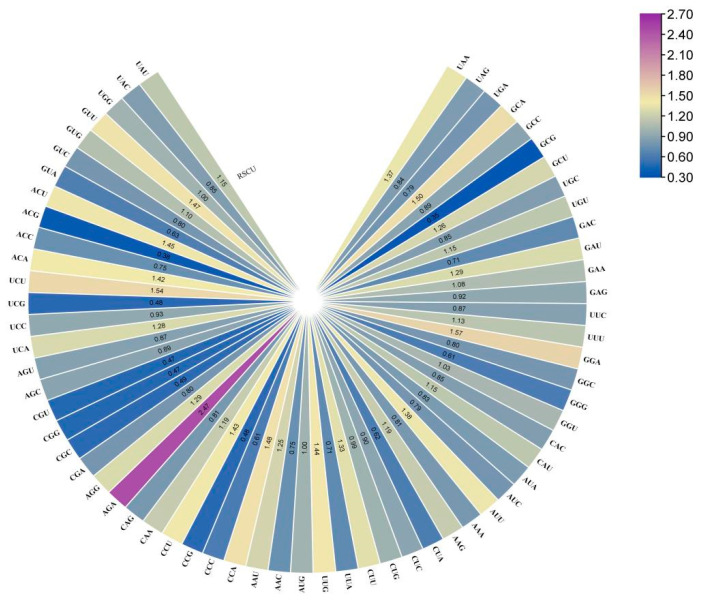
RSCU heat map analysis of the protein coding sequence of the strawberry *DFR* genes. The deep purple indicates a codon preference.

**Figure 7 ijms-25-09911-f007:**
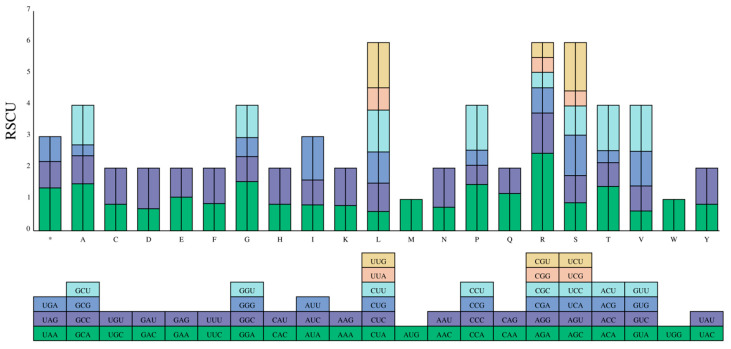
RSCU analysis of the protein-encoding sequence of the strawberry *DFR* gene.

**Figure 8 ijms-25-09911-f008:**
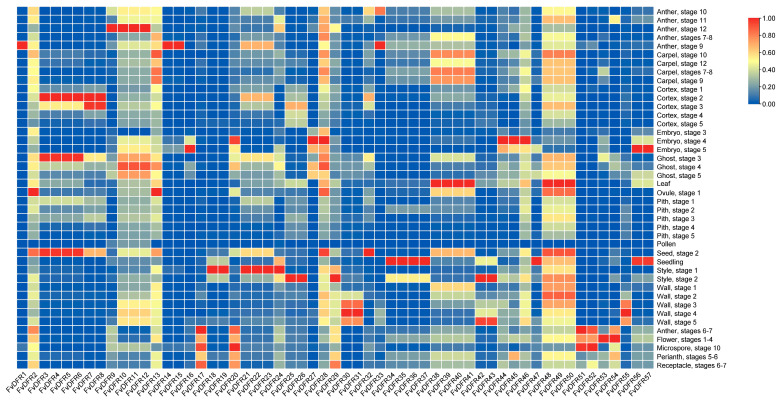
Expression of strawberry *DFR* in different tissues. Shades of red and blue represent up-regulated or down-regulated expression levels, respectively. Metrics represent relative expression levels.

**Figure 9 ijms-25-09911-f009:**
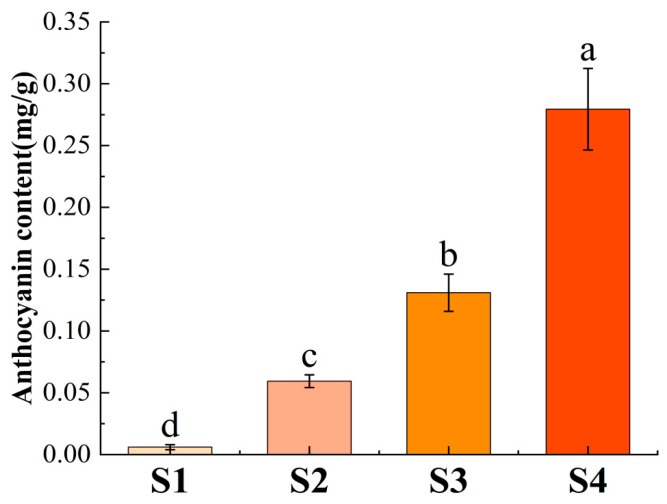
The strawberry fruit used in this study is fresh fruit, calculated by fresh weight. The anthocyanin content in the four periods. S1 is the green fruit stage, S2 is the 20% colored period, S3 is the 50% colored period, and S4 is the completely colored period. These data are original to this manuscript. Different letters denote significant differences (*p* < 0.05).

**Figure 10 ijms-25-09911-f010:**
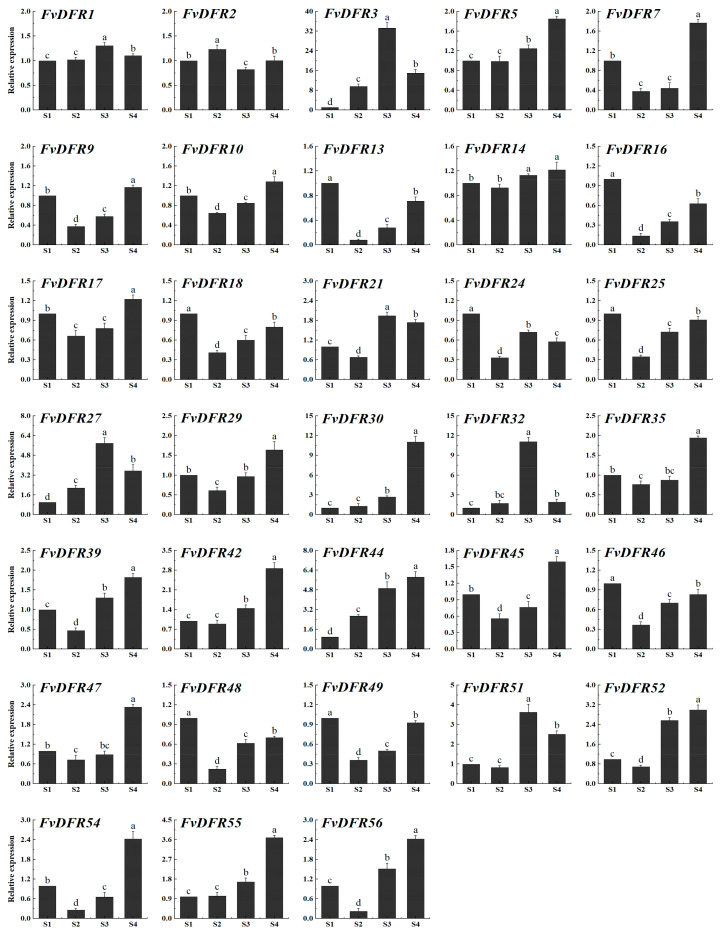
Quantitative expression analysis of the *DFR* gene family in strawberry. Strawberry fruits at the green fruit stage (S1), 20% coloring stage (S2), 50% coloring stage (S3), and completely colored stage (S4) were selected, and S1 was used as the control. There were 3 replicates per treatment, and the reference gene was *GAPDH*. Different lowercase letters indicate a significant difference at the 0.05 level, and the same lowercase letters indicate no statistical difference. According to the six subgroups of the strawberry *DFR* gene family and their evolutionary relationship, there are similarities in the same class during classification, so different genes were selected from the different subgroups when conducting research, so some genes were excluded. These data are original data from this paper.

**Figure 11 ijms-25-09911-f011:**
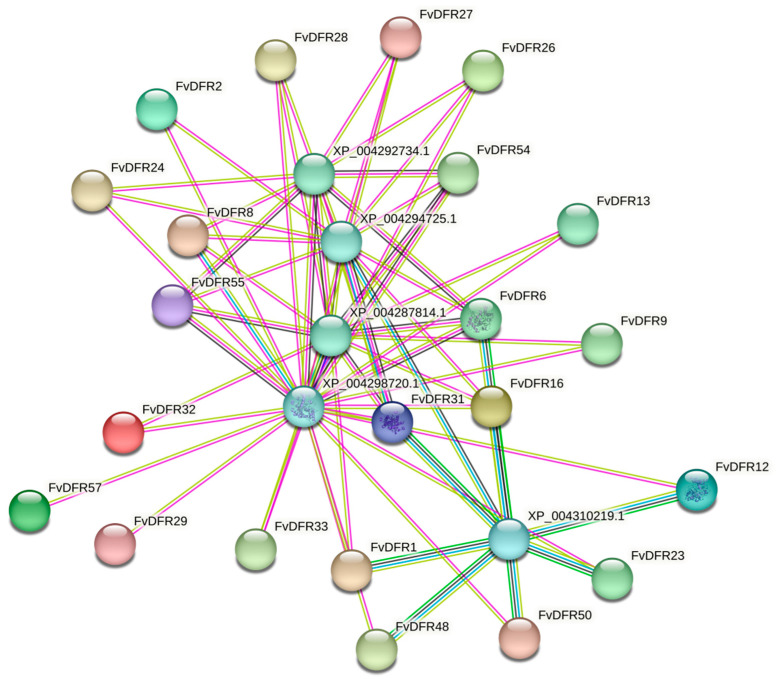
Protein interaction analysis of the strawberry *DFR* gene family. Nodes represent proteins, and lines between nodes represent interactions between proteins, with different colors corresponding to different types of interactions.

**Table 1 ijms-25-09911-t001:** Physicochemical properties of the strawberry *DFR* gene family, including number of amino acids, molecular weight/Da, pI, instability index, aliphatic index, and grand average of hydropathicity.

Gene Name	GenBank No.	Number of Amino Acids	Molecular Weight/Da	pI	Instability Index	Aliphatic Index	Grand Average of Hydropathicity
*FvDFR1*	FvH4_2g17230.t1.v4.0.a2	329	36,554.02	6.27	39.77	97.48	−0.145
*FvDFR2*	FvH4_2g34590.t1.v4.0.a2	336	37,093.61	7.74	31.35	95.42	−0.026
*FvDFR3*	FvH4_2g39520.t1.v4.0.a2	349	38,956.7	6.03	36.65	82.64	−0.223
*FvDFR4*	FvH4_2g39520.t2.v4.0.a2	281	31,308.21	6.75	35.3	90.57	−0.067
*FvDFR5*	FvH4_2g39520.t5.v4.0.a2	331	37,150.87	6.28	38.34	86.31	−0.162
*FvDFR6*	FvH4_2g39520.t3.v4.0.a2	333	37,202.99	6.43	37.21	87.57	−0.126
*FvDFR7*	FvH4_3g02980.t2.v4.0.a2	275	29,731.37	5.43	23.07	99.96	0.194
*FvDFR8*	FvH4_3g02980.t1.v4.0.a2	341	36,861.42	6.25	28.09	93.49	−0.012
*FvDFR9*	FvH4_3g21610.t1.v4.0.a2	354	37,780.97	5.1	44.75	97.23	0.063
*FvDFR10*	FvH4_3g28560.t2.v4.0.a2	285	31,371.08	7.62	28.01	92.67	−0.081
*FvDFR11*	FvH4_3g28560.t1.v4.0.a2	289	31,962.82	8.39	27.1	91.38	−0.076
*FvDFR12*	FvH4_3g28560.t3.v4.0.a2	321	35,294.56	6.19	25.32	92.27	−0.062
*FvDFR13*	FvH4_3g39460.t1.v4.0.a2	324	35,438.78	5.68	28.15	96.85	−0.01
*FvDFR14*	FvH4_4g27430.t1.v4.0.a2	319	35,199.2	5.41	37.96	88.65	−0.087
*FvDFR15*	FvH4_4g27430.t2.v4.0.a2	229	25,523.15	5.87	44.24	83.93	−0.141
*FvDFR16*	FvH4_4g27470.t1.v4.0.a2	69	7686.98	9.43	24.09	93.33	−0.013
*FvDFR17*	FvH4_5g01060.t1.v4.0.a2	340	36,937.43	8.6	38.59	89.15	−0.207
*FvDFR18*	FvH4_5g04220.t2.v4.0.a2	296	32,710.5	5.63	42.37	87.94	−0.033
*FvDFR19*	FvH4_5g04220.t1.v4.0.a2	345	38,216.82	5.68	42.72	86.17	−0.105
*FvDFR20*	FvH4_5g04260.t1.v4.0.a2	295	32,503.36	5.49	41.21	90.88	0.003
*FvDFR21*	FvH4_5g22420.t2.v4.0.a2	276	30,947.39	5.2	33.71	87.93	−0.113
*FvDFR22*	FvH4_5g22420.t3.v4.0.a2	208	23,072.49	6.42	26.94	91.39	−0.075
*FvDFR23*	FvH4_5g22420.t1.v4.0.a2	315	34,963.73	5.8	37.03	83.87	−0.182
*FvDFR24*	FvH4_5g26020.t1.v4.0.a2	319	35,559.9	6.9	28.47	95.61	−0.213
*FvDFR25*	FvH4_5g26030.t2.v4.0.a2	298	33,155.86	5.26	34.07	89.9	−0.208
*FvDFR26*	FvH4_5g26030.t1.v4.0.a2	326	36,038.11	5.47	31.69	87.85	−0.206
*FvDFR27*	FvH4_5g26090.t1.v4.0.a2	325	36,066.5	7.47	35.87	87.51	−0.231
*FvDFR28*	FvH4_5g27181.t1.v4.0.a2	331	35,783.64	5.76	38.84	87.79	−0.063
*FvDFR29*	FvH4_5g33950.t1.v4.0.a2	300	33,330.22	6.26	41.39	84.2	−0.172
*FvDFR30*	FvH4_6g28680.t1.v4.0.a2	339	37,193.48	6.17	34.23	92.57	−0.206
*FvDFR31*	FvH4_6g28680.t2.v4.0.a2	279	30,321.63	5.78	32.07	95.02	−0.045
*FvDFR32*	FvH4_6g45521.t1.v4.0.a2	347	38,638.31	5.92	35.14	86.83	−0.178
*FvDFR33*	FvH4_6g50850.t1.v4.0.a2	356	39,502.16	7.53	38.3	90.08	−0.196
*FvDFR34*	FvH4_7g01511.t3.v4.0.a2	315	34,545.75	5.1	22.84	98.7	0.003
*FvDFR35*	FvH4_7g01511.t2.v4.0.a2	233	25,752.74	5.24	29.24	96.61	0.083
*FvDFR36*	FvH4_7g01511.t4.v4.0.a2	109	11,760.52	5.63	14.46	100.09	0.117
*FvDFR37*	FvH4_7g01511.t1.v4.0.a2	325	35,637.98	5.03	23.63	97.75	0.01
*FvDFR38*	FvH4_7g01520.t3.v4.0.a2	292	32,217	5.3	36.75	100.17	−0.047
*FvDFR39*	FvH4_7g01520.t2.v4.0.a2	229	24,937.86	6.9	23.09	105.94	0.128
*FvDFR40*	FvH4_7g01520.t4.v4.0.a2	334	37,337.49	8.75	29.31	108.86	0.095
*FvDFR41*	FvH4_7g01520.t1.v4.0.a2	328	35,944.43	5.93	30.99	102.53	0.013
*FvDFR42*	FvH4_7g01560.t2.v4.0.a2	297	32,874.99	5.87	33.82	98.11	0.01
*FvDFR43*	FvH4_7g01560.t1.v4.0.a2	328	35,963.57	6.51	30.53	99.24	0.03
*FvDFR44*	FvH4_7g01561.t1.v4.0.a2	200	22,086.52	5.13	27.98	102.4	0.049
*FvDFR45*	FvH4_7g01562.t1.v4.0.a2	113	12,170.15	8.4	30.5	103.45	0.249
*FvDFR46*	FvH4_7g01563.t1.v4.0.a2	223	24,257.09	6.43	27.89	98.74	0.07
*FvDFR47*	FvH4_7g01564.t1.v4.0.a2	323	35,549.88	5.28	25.65	96.22	0.021
*FvDFR48*	FvH4_7g01580.t3.v4.0.a2	322	35,390.69	6.14	29.12	95.68	−0.039
*FvDFR49*	FvH4_7g01580.t2.v4.0.a2	1253	136,295.17	5.85	32.8	101.35	0.056
*FvDFR50*	FvH4_7g01580.t1.v4.0.a2	344	37,808.72	8.61	27.73	96.34	−0.071
*FvDFR51*	FvH4_7g11750.t1.v4.0.a2	126	14,089.26	4.69	33.68	104.37	0.185
*FvDFR52*	FvH4_7g11760.t1.v4.0.a2	296	32,516.22	6.32	35.39	85.61	−0.176
*FvDFR53*	FvH4_7g11780.t1.v4.0.a2	334	36,985.41	6.08	39.18	85.81	−0.126
*FvDFR54*	FvH4_7g11781.t1.v4.0.a2	328	36,100.34	5.64	39.55	92.1	−0.018
*FvDFR55*	FvH4_7g11790.t1.v4.0.a2	331	36,612.02	5.25	32.18	87.13	−0.05
*FvDFR56*	FvH4_7g16111.t2.v4.0.a2	313	34,918.03	6.51	46.13	88.43	−0.105
*FvDFR57*	FvH4_7g16111.t1.v4.0.a2	357	39,939.81	6.32	45.53	87.37	−0.128

**Table 2 ijms-25-09911-t002:** Subcellular localization prediction of the strawberry *DFR* gene family.

Protein	Cytoplasm	Chloroplast	Nucleus	Plasma Membrane	Mitochondria	Peroxisome
*FvDFR1*	3	4	3	-	1	-
*FvDFR2*	-	11.5	-	-	2.5	-
*FvDFR3*	4	4	1	2	-	-
*FvDFR4*	1	9	-	2	-	-
*FvDFR5*	4	4	2	2	-	-
*FvDFR6*	1	8	1	2	-	-
*FvDFR7*	4.5	1	1.5	-	1	-
*FvDFR8*	2	7	1	-	1	-
*FvDFR9*	10	-	2	-	1	-
*FvDFR10*	2	8	-	1	1	-
*FvDFR11*	1	10	-	1	1	-
*FvDFR12*	1	9	-	1	1	-
*FvDFR13*	3	9	1	-	-	-
*FvDFR14*	8	2	1	-	-	-
*FvDFR15*	-	10	1	-	3	-
*FvDFR16*	11.5	-	1	-	-	-
*FvDFR17*	8	1	-	-	1	-
*FvDFR18*	2	10	2	-	-	-
*FvDFR19*	1	11	2	-	-	-
*FvDFR20*	-	12	1	-	1	-
*FvDFR21*	10	2	-	1	-	-
*FvDFR22*	4	9	-	-	-	1
*FvDFR23*	9	2	2	1	-	-
*FvDFR24*	2	7	1	-	-	-
*FvDFR25*	8	3	-	-	1	2
*FvDFR26*	9	3	-	-	-	2
*FvDFR27*	11	1	1	-	-	1
*FvDFR28*	4	8	-	-	-	-
*FvDFR29*	3	2	7	-	1	-
*FvDFR30*	2	11	-	-	-	-
*FvDFR31*	1	10	-	1	1	-
*FvDFR32*	11	1	1	1	-	-
*FvDFR33*	1	4	2	2	-	-
*FvDFR34*	11	-	-	-	-	2
*FvDFR35*	9.5	3	1	-	-	-
*FvDFR36*	12	1	1	-	-	-
*FvDFR37*	9	1	1	-	-	2
*FvDFR38*	4	2	-	-	-	-
*FvDFR39*	1	9	1	-	-	-
*FvDFR40*	3	10	1	-	-	-
*FvDFR41*	2	8	1	-	-	-
*FvDFR42*	6.5	1	1.5	-	2	-
*FvDFR43*	6	7	-	1	-	-
*FvDFR44*	10	2	-	-	-	-
*FvDFR45*	2	10	-	-	1	-
*FvDFR46*	-	14	-	-	-	-
*FvDFR47*	3	9	1	-	-	-
*FvDFR48*	-	9	-	2	-	-
*FvDFR49*	1	2	1	9	-	-
*FvDFR50*	1.5	4	3.5	-	3	1
*FvDFR51*	7	-	1	1.5	-	1
*FvDFR52*	5	5	1	2	-	-
*FvDFR53*	3	4	2	1	2	-
*FvDFR54*	7	1	-	-	1	-
*FvDFR55*	2	4	1	2	-	-
*FvDFR56*	1	5.5	3	-	4	-
*FvDFR57*	1	3.5	3	1	5	-

**Table 3 ijms-25-09911-t003:** Secondary structure analysis of the *DFR* gene family proteins in strawberry, including alpha helices, extended strands, beta turns, and random coils.

Protein	Alpha Helix/%	Extended Strand/%	Beta Turn/%	Random Coil/%
*FvDFR1*	42.86	13.07	7.29	36.78
*FvDFR2*	45.24	13.69	7.14	33.93
*FvDFR3*	38.97	14.33	7.16	39.54
*FvDFR4*	42.35	16.37	6.76	34.52
*FvDFR5*	38.97	13.90	8.16	38.97
*FvDFR6*	41.44	14.11	7.81	36.64
*FvDFR7*	46.18	14.55	5.09	34.18
*FvDFR8*	41.64	14.96	6.74	36.66
*FvDFR9*	40.68	13.56	5.93	39.83
*FvDFR10*	39.65	16.49	8.07	35.79
*FvDFR11*	39.79	16.96	8.30	34.95
*FvDFR12*	42.06	16.82	7.17	33.96
*FvDFR13*	45.68	14.81	7.10	32.41
*FvDFR14*	41.69	14.42	7.52	36.36
*FvDFR15*	41.48	13.54	4.80	40.17
*FvDFR16*	23.19	31.88	7.25	37.68
*FvDFR17*	42.94	17.35	5.00	34.71
*FvDFR18*	40.20	14.53	7.77	37.50
*FvDFR19*	40.87	13.91	7.25	37.97
*FvDFR20*	38.98	16.61	7.12	37.29
*FvDFR21*	44.93	16.67	5.43	32.97
*FvDFR22*	46.63	17.31	6.73	29.33
*FvDFR23*	44.44	17.14	4.13	34.29
*FvDFR24*	46.71	12.54	7.21	33.54
*FvDFR25*	48.99	12.08	4.36	34.56
*FvDFR26*	44.79	13.80	6.13	35.28
*FvDFR27*	46.77	14.15	5.54	33.54
*FvDFR28*	43.81	12.39	6.95	36.86
*FvDFR29*	38.00	15.00	8.33	38.67
*FvDFR30*	41.89	14.75	6.78	36.58
*FvDFR31*	43.37	16.49	6.45	33.69
*FvDFR32*	51.59	13.26	6.05	29.11
*FvDFR33*	49.16	11.80	4.49	34.55
*FvDFR34*	42.86	16.83	6.03	34.29
*FvDFR35*	41.63	17.17	6.44	34.76
*FvDFR36*	41.28	17.43	11.01	30.28
*FvDFR37*	42.15	15.38	5.85	36.62
*FvDFR38*	41.10	14.38	6.16	38.36
*FvDFR39*	42.79	17.03	6.99	33.19
*FvDFR40*	41.02	18.86	6.89	33.23
*FvDFR41*	41.16	14.63	6.10	38.11
*FvDFR42*	41.75	16.84	6.73	34.68
*FvDFR43*	42.07	16.77	7.01	34.15
*FvDFR44*	45.00	14.50	5.50	35.00
*FvDFR45*	46.90	18.58	10.62	23.89
*FvDFR46*	42.60	17.94	7.17	32.29
*FvDFR47*	43.96	16.10	7.74	32.20
*FvDFR48*	43.17	17.39	6.83	32.61
*FvDFR49*	34.80	17.64	3.27	44.29
*FvDFR50*	43.02	15.41	5.52	36.05
*FvDFR51*	36.51	29.37	8.73	25.40
*FvDFR52*	42.91	15.88	7.77	33.45
*FvDFR53*	41.62	14.97	8.98	34.43
*FvDFR54*	42.99	15.55	7.32	34.15
*FvDFR55*	37.76	14.50	7.55	40.18
*FvDFR56*	40.26	15.65	7.67	36.42
*FvDFR57*	40.62	14.57	7.28	37.54

**Table 4 ijms-25-09911-t004:** Amino acid-conserved sequences of the strawberry *DFR* gene family.

Conserved Motif	Length/bp	Amino Acid-Conserved Sequence
Motif1	31	RLHLFKADLLDEGSFDSAIDGCEGVFHTASP
Motif2	38	SKTLAEQAAWKFAKENGJDLVTINPGLVIGPLLQPTLN
Motif3	32	VVCVTGASGFIASWLVKLLLZRGYTVKATVRB
Motif4	26	YRFVHVDDVASAHIFAFENPSASGRY
Motif5	21	DPZAELIDPAVKGTLNVLKSC
Motif6	21	KSPTVKRVVLTSSIAAVAYNG
Motif7	15	WWSDPEFCEKLKLWY
Motif8	21	HISEIVKILRELYPEYNIPEK
Motif9	15	NDLKKTEHLLSLDGA
Motif10	29	QVSKEKAQSLGVKFTPLEVTLKDTVESLK

**Table 5 ijms-25-09911-t005:** Primers for real-time quantification of the strawberry *DFR* gene family.

Gene	Forward Primer	Reverse Primer
*GAPDH*	CATTCATCACCACCGACTACA	GAAGGGTCTTCTCATCCTTGAC
*FvDFR1*	CATGTCACTGGAACTGTCAGAGATC	TCAACTAAGTCAGCCTTCACCAATC
*FvDFR2*	CACGCCGCCGCCTTAGTC	CTTGGTCTCCCTGATCGCTCTC
*FvDFR3*	GCTGAGCAAGAGGCATGGAAG	TGAGACTCGGCGGCATAGC
*FvDFR5*	GGACCCTGAGAACGAAGTGATAAAG	CGATGCTCTTCAATGGCGACAG
*FvDFR7*	ACCACTGCCAAGCCACCTAC	AGAGAAGCACCAGCCATGAGAG
*FvDFR9*	CCACACGGTTCATGCCACAG	AAACGCCATCACAGCCCTTAAC
*FvDFR10*	CGCAACCCAAATGATCCAACAAAG	AACACATTCACAGCCCTCAACAC
*FvDFR13*	CGTCTTCCACCTCGCCTCTC	GCCGCCGTCAGAACATTCAG
*FvDFR14*	CGGAGAAAGATGCGTGGAGAATC	GTGCTTGTAGGTTGTGGTGCTAG
*FvDFR16*	CGAGACCCAGCCATAGCCTTC	CCTTGGTGTGCATGAAACTTCCC
*FvDFR17*	ATATTTGTGGTGGTAGGAGGAGGAG	TCGTCGGTGGCTATGTCTGTG
*FvDFR18*	ACACAGTCCATGCCACTCTCAG	AGCAGGCTTGAAGTCATTAGGATTG
*FvDFR21*	GCGTGGGCAATGGCAATGG	GATGGACAAGTCGGCGTTCAAC
*FvDFR24*	AGGTGCTGGAGGCTTCGTTG	TCTCTGATGGTTCCGTGGACTATG
*FvDFR25*	GCAATCCACCCTAATTTCCAGTAGC	TCACGAACATCCACCACAGTCC
*FvDFR27*	AACTTGGTGCTGATGTCTCTTCTG	TGTATCTGCCTCCTGCTTCTGG
*FvDFR29*	GTTGGACACCGCTTGATGCC	CCCAGTAAGTTGTGCCAGAATCC
*FvDFR30*	GAGAGGAGTGGACTTGGTGGTG	GGATGATGCTGGCGTTGATGG
*FvDFR32*	GGAAGACCACTGCGGAGATAGC	CATGTCCTTGCCACCTTTGAGATAG
*FvDFR35*	ACAGAACACTTGCTCGCACTTG	CACCTTCACATCCATCAACAACAAG
*FvDFR39*	AAACAGAACACTTGCTTTCGCTTG	AACACCATCACATCCATCAACCG
*FvDFR42*	CGCACATCTCCGAGGCTCTG	GGTCAGAAGGAAGAATACGAACAGG
*FvDFR44*	TCTTACAGCCAACTCTGAACACAAG	CGCAATCGCAACATCTCTAACATC
*FvDFR45*	TGTGACAGGAGCATCTGGGTTC	CGGACAGTGGCTTTGACAGTATAAC
*FvDFR46*	CACCTGAGACACCTGTAATAGTTCC	GCTGCCATAGAGGATGTGATAACC
*FvDFR47*	CAGAACACCTGCTCTCACTTGAAG	AGAAGTACAGAGGATGCCGTATGG
*FvDFR48*	TGAACTAATTGACCCTGCCTTGAAG	ACCACCCTCTTAATAGACGGAACC
*FvDFR49*	GTTCAATCGGGTCTCCTTTGGTTC	TGGCAGCAGTCACTTCCTTCC
*FvDFR51*	GCAAGTCAAAGCGGAATTTCAGATG	ATCGTCGCCACCACCAAGG
*FvDFR52*	CTGAGAGCCTGCCTGAAGTCC	ATGTCTTTGTTGGTTCCGCTATGAG
*FvDFR54*	TGCGAGCCACCGTTAGATCC	TCATTGAAGGTCTCAGGGTTGTTG
*FvDFR55*	GAGGTTCTGGGTTCATTGGTTCC	CTTTCTTGTCTTCTGCTGGGTCTG
*FvDFR56*	TTCAAAGACACTGGCAGAGAAAGAG	ATTCCCTCCCACAAGACCACAG

## Data Availability

The data that support the finding of this study are available from the corresponding author upon reasonable request.

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
