# Peer review of "The Genome-Wide Identification of the Dihydroflavonol 4-Reductase (DFR) Gene Family and Its Expression Analysis in Different Fruit Coloring Stages of Strawberry"

_ijms, 2024, doi:10.3390/ijms25189911_

Round 1

Reviewer 1 Report

Comments and Suggestions for Authors

The manuscript describes the members of the DFR gene family strawberry and their expression in relation to anthocyanin concentrations in fruit.  They describe 57 members of the gene family and find that four members (FvDRF3, FvDRF27, FvDRF32, FvDRF51) had highest expression in the rapid fruit coloring stage while three members (FvDRF30, FvDRF54, and FvDRF56) had highest expression in the complete fruit coloring stage. While the study potentially provides useful information about important set of genes in an important crop species, the manuscript has several significant problems in the current form.

The major issues are as follows. 

1.      The first paragraph of the discussion is not coherent or acceptable in the current form. “And others in a variety of plants.” is not a complete sentence. Swallow flower is not a well-known plant reference. Comparison should be restricted to well known taxa to readers. Finally, one statement is totally incorrect. “Compared with 55 members of rice there was little difference in numbers…which may be due to a relatively close relationship”  -  this sentence is saying that strawberry (a rosid) has a close relationship to rice (a grass).  Please correct this.  Both the rosids and the grasses have had several independent WGD events that increased the total genomes of each lineage 8x or so.

2.      Please describe the units in Table 3 regarding subcellular localization in the table title.  It is not clear what the numbers refer to here. 

3.      In Figure 3, the evolutionary analysis of DFR gene families is used to divide the genes into six families (A-F). Please add the bootstrapping support for each of these groups.  To the reader it appears clear that there is strong support for D, E, and F, but families A and B each have one member and do not appear resolved relative to Family C.  Currently, the tree does not provide sufficient evidence to support the distinctions made in the text. 

4.      Please make it clearer that Fig 8 is expression data mined from online data that are provided in Tbtools. The methods section should cite the actual study that was the source of the expression dataset.

5.      Please add more to the legend of Fig.9.  Clarify to the reader that these data are original to this manuscript.  Please describe in a sentence or so the sampling information. Please clarify whether this is per fresh weight or dry weight. 

6.      Please clarify in Fig 10 legend why some of the genes are excluded. What was the basis for the inclusion/ exclusion.  Please clarify how these data show that four members (FvDRF3, FvDRF27, FvDRF32, FvDRF51) had highest expression in the rapid fruit coloring stage while three members (FvDRF30, FvDRF54, and FvDRF56) had highest expression in the complete fruit coloring stage. Clarify also in the legend that these data are original to the paper.

7.      The protein interaction plot in Fig 11 is interesting but please link the gene IDs (XP_004292734.1, for example) with the names (4-hydroxycoumarin synthase, for example) where they are mentioned in the text or alternatively in the figure legend.

8.      Finally, the references have many numbering errors in the text which made the manuscript difficult to read and must be corrected.  Some of these references are to thesis documents at universities.  It is generally not useful to cite theses if it can be avoided.

Comments on the Quality of English Language

The English is generally okay, except as noted

Author Response

Dear Editors and Reviewers,

Thank you for your letter and for the comments concerning our manuscript titled “Genome wide Identification of Dihydroflavonol 4-reductase (DFR) gene family and its expression analysis in different coloring stages of fruit in strawberry”. The comments have helped us improve our manuscript and have been valuable for our future study. We have studied the comments carefully and have applied corrections, which we hope would meet your approval. The main corrections in the paper and the response to the comments are as follows:

Comment: The first paragraph of the discussion is not coherent or acceptable in the current form. “And others in a variety of plants.” is not a complete sentence. Swallow flower is not a well-known plant reference. Comparison should be restricted to well known taxa to readers. Finally, one statement is totally incorrect. “Compared with 55 members of rice there was little difference in numbers…which may be due to a relatively close relationship” - this sentence is saying that strawberry (a rosid) has a close relationship to rice (a grass).  Please correct this. Both the rosids and the grasses have had several independent WGD events that increased the total genomes of each lineage 8x or so.

Response: Thank you for your valuable suggestion. “And others in a variety of plants.” has been changed “And others are in a variety of plants.” Swallow flower has been changed to Brassica rapa. The text has been changed “In this study, 57 members of DFR gene family were identified in strawberry genome database by bioinformatics method. However, compared with the 26 members of rape and the 96 members of apple, the number of DFR gene families in strawberry is larger, reflecting the large differences between different species.”(line291-294)

Comment: Please describe the units in Table 3 regarding subcellular localization in the table title.  It is not clear what the numbers refer to here.

Response: Thank you for your valuable suggestion. Units have been added to the table header of Table 3.

Comment: In Figure 3, the evolutionary analysis of DFR gene families is used to divide the genes into six families (A-F). Please add the bootstrapping support for each of these groups.  To the reader it appears clear that there is strong support for D, E, and F, but families A and B each have one member and do not appear resolved relative to Family C. Currently, the tree does not provide sufficient evidence to support the distinctions made in the text.

Response: Thank you for your valuable suggestion. We searched for published research papers on the DFR gene family and found that most crops were divided into three subfamilies, and this study also referred to these findings to classify the strawberry DFR family, and we can also divide the strawberry DFR family into three subgroups. However, from the classification results, the three subgroups contained 1, 6 and 50 members, respectively, and one of the subfamilies contained the vast majority of family members, which was not conducive to the study of gene structure and real-time fluorescence quantitative expression analysis on gene function prediction in the later stage, and could not distinguish the conservatism of family member function well. Therefore, we combined the clustering results and classified the family members into 6 subfamilies by showing the position classification in the figure below, which is convenient for later research.

Comment: Please make it clearer that Fig 8 is expression data mined from online data that are provided in Tbtools. The methods section should cite the actual study that was the source of the expression dataset.

Response: Thank you for your valuable suggestion. The data was mined from the BAR (https://bar.utoronto.ca/) and visualized using TBtools.

Comment: Please add more to the legend of Fig.9. Clarify to the reader that these data are original to this manuscript.  Please describe in a sentence or so the sampling information. Please clarify whether this is per fresh weight or dry weight.

Response: Thank you for your valuable suggestion. The sampling information has been explained in the legend of Figure 9: the strawberry fruit used in this study is fresh fruit, calculated by fresh weight. And it has been explained to readers that these data are the original data of this manuscript.(line250-253)

Comment: Please clarify in Fig 10 legend why some of the genes are excluded. What was the basis for the inclusion/ exclusion. Please clarify how these data show that four members (FvDRF3, FvDRF27, FvDRF32, FvDRF51) had highest expression in the rapid fruit coloring stage while three members (FvDRF30, FvDRF54, and FvDRF56) had highest expression in the complete fruit coloring stage. Clarify also in the legend that these data are original to the paper.

Response: Thank you for your valuable suggestion. In this study, strawberry DFR gene family is divided into six subfamilies, which are classified according to different subfamilies and their evolutionary relationships, there are similarities in the same category. So some genes are excluded when selecting genes. FvDFR3, FvDFR27, FvDFR32, and FvDFR51 had the highest gene expression levels in the S3 phase. Overall, expression levels increased first and then decreased. This may be due to the high expression of this gene during the rapid accumulation of fruit pigments and may regulate anthocyanin synthesis. FvDFR30, FvDFR54 and FvDFR56 were the highest in S4 phase. It may be due to the cumulative effect of expression of these three genes or the sluggish response, which may cause a component to accumulate rapidly despite the completion of coloring; It may also be due to the fact that these three genes have a conservation effect on color, so that the color-related components are not degraded after the fruit coloring is completed. The authors have explained in the legend that these data are original to the paper.

Comment: The protein interaction plot in Fig 11 is interesting but please link the gene IDs (XP_004292734.1, for example) with the names (4-hydroxycoumarin synthase, for example) where they are mentioned in the text or alternatively in the figure legend.

Response: Thank you for your valuable suggestion. Because protein-protein interaction mapping is difficult to redo, the authors have linked gene IDs (e.g., XP_004292734.1) to names (e.g., 4-hydroxycoumarin synthase) in the text.

Comment: Finally, the references have many numbering errors in the text which made the manuscript difficult to read and must be corrected. Some of these references are to thesis documents at universities.  It is generally not useful to cite theses if it can be avoided.

Response: Thank you for your valuable suggestion. References and numbering have been changed.

Reviewer 2 Report

Comments and Suggestions for Authors

Dear authors,

In the present study, the authors have identified and characterized 57 members of the strawberry DFR gene family and then analyzed their expression in different coloring stages of fruit (from S1 to S4). Similar studies for several anthocyanin biosynthesis-related gene families in Strawberry have been published by author(s)’ group: for ANS family: https://www.mdpi.com/1422-0067/24/16/12554 and https://doi.org/10.1186/s12870-023-04648-3 and for F3H: https://www.mdpi.com/1422-0067/24/23/16807. Herein, using a similar approach the authors conducted another gene, DFR, one of the serial genes of anthocyanin biosynthesis pathway.

Comments and Suggestions for Authors

- Why did author(s) select DFR and other genes (ANS, and F3H) for study? If the authors have any preliminary findings, such as that these gene expressions are rate-limiting in anthocyanin biosynthesis. Does the author(s) plan to study all remaining genes of anthocyanin biosynthesis pathways such as PAL, CHS, CHI, and UFGT…?

- Named/designated four periods (S1-S4) as [S1 is the green fruit stage, S2 is 20% colored, S3 is 50% colored period, respectively, and S4 is the complete colored] should be described in the section “Plant materials” (line 330).

- Why did the authors analyze the expression (qRT-PCR) of only 33 DFR genes among 57 DFR gene families? As Fig 10, some genes such as FvDFR4, 6,8,11,12,.. were not analyzed.

- Please update the citation. There was a citation error that happened showing as "Error! Reference source not found". It started with a citation of Ref 11 (line 68). In addition, the citation format (cited number) should follow the journal's instructions.

- This sentence should be rewritten: “In this study, the expression of the gene family members was studied by strawberry quantitative PCR.” (line 85-86) since “strawberry quantitative PCR” is an unclear meaning.

- Relabeled/renumbered the table. The table 2 should be table 1. I guess this error may have occurred during the section order of the IJMS journal in which section “4. Materials and Methods” was put behind the section “2. Results”. Shift for all tables and do not forget to cite with an update once in the text. For example, the cited Table 2 (line 93) should be cited by Table 1, and Table 1 (line 401, 393) should be changed to Table 5. Please carefully double-check in whole manuscript.

- Some Figs (Fig 2, 8) presented in the manuscript were of low quality. The authors should replace it with a new one of high quality. Tip, the Fig data might be placed in the text of the manuscript in high resolution by choosing the paste option as insert the contents of clipboard as files, do not paste as pictures.

- Represent the manuscript according to the journal’s style:

+ Section “Abstract” should be removed sentence/paragraph section head (Background, Methods, Results, and Conclusions) since it was presented following other journals’ style (Biomedcentral, BMC).

+ Numbered the subsections and Capitalize Each Word in the paper title and subsection.

For example, subsection “Sequence characteristics of the strawberry DFR gene family” (line 90) -> “2.1. Sequence Characteristics of the Strawberry DFR Gene Family”

Minor remarks:

- Italicize gene name: DFR (line 77, 78, ect)

- Italicize the scientific name of plant: Nicotiana gossei, Arabidopsis thaliana (Fig 3’s legend)

- Change the unit of (protein molecular weight) from “D” (line 95) to “Da.

- Lowercase “Alpha helix”, “Random coil”, “Extended strand”, and “Beta turn” -> “alpha helix”, “random coil”, “extended strand”, and “beta turn”.

-  Grammar error: “Arabidopsis Thaliana” (line 430) change to “Arabidopsis thaliana

I have marked the above comments on the manuscript. Please check.

Author Response

Dear Editors and Reviewers,

Thank you for your letter and for the comments concerning our manuscript titled “Genome wide Identification of Dihydroflavonol 4-reductase (DFR) gene family and its expression analysis in different coloring stages of fruit in strawberry”. The comments have helped us improve our manuscript and have been valuable for our future study. We have studied the comments carefully and have applied corrections, which we hope would meet your approval. The main corrections in the paper and the response to the comments are as follows:

Comment: Why did author(s) select DFR and other genes (ANS, and F3H) for study? If the authors have any preliminary findings, such as that these gene expressions are rate-limiting in anthocyanin biosynthesis. Does the author(s) plan to study all remaining genes of anthocyanin biosynthesis pathways such as PAL, CHS, CHI, and UFGT…?

Response:  Thank you for your valuable suggestion. As you said, there are many functional genes and transcription factors involved in the regulation of anthocyanin synthesis pathway, we have cloned and analyzed the ANS and F3H gene families of strawberry in the early stage, preliminarily clarified the expression level of each member in the process of fruit coloring, screened out the key genes involved in color regulation in different fruit coloring stages, and combined with yeast hybridization technology, the regulatory module involved in anthocyanin synthesis was preliminarily screened, enriching the anthocyanin metabolism pathway. There are no plans to carry out relevant research on other functional genes in the anthocyanin synthesis pathway, such as PAL, CHS, CHI, and UFGT, focusing on the cascade regulatory modules of key candidate gene functions obtained in ANS and F3H family members, and then planning to start with other functional genes.

Comment: Named/designated four periods (S1-S4) as [S1 is the green fruit stage, S2 is 20% colored, S3 is 50% colored period, respectively, and S4 is the complete colored] should be described in the section “Plant materials” (line 330).

Response: Thank you for your valuable suggestion. Four named/designated periods (S1-S4) have been described in the section “Plant materials” (line352).

Comment: Why did the authors analyze the expression (qRT-PCR) of only 33 DFR genes among 57 DFR gene families? As Fig 10, some genes such as FvDFR4, 6,8,11,12,.. were not analyzed.

Response: Thank you for your valuable suggestion. In this study, strawberry DFR gene family is divided into six subfamilies, which are classified according to different subfamilies and their evolutionary relationships, there are similarities in the same category. So the authors analyze the expression (qRT-PCR) of only 33 DFR genes among 57 DFR gene families.

Comment: Please update the citation. There was a citation error that happened showing as "Error! Reference source not found". It started with a citation of Ref 11 (line 68). In addition, the citation format (cited number) should follow the journal's instructions.

Response: Thank you for your valuable suggestion. The quotation in the article has been updated. Citation formatting has changed.

Comment: This sentence should be rewritten: “In this study, the expression of the gene family members was studied by strawberry quantitative PCR.” since “strawberry quantitative PCR” is an unclear meaning.

Response: Thank you for your valuable suggestion. This sentence has been rewritten to study the expression of DFR gene family in strawberry by real-time fluorescence quantitative PCR (line84-85).

Comment: Relabeled/renumbered the table. The table 2 should be table 1. I guess this error may have occurred during the section order of the IJMS journal in which section “4. Materials and Methods” was put behind the section “2. Results”. Shift for all tables and do not forget to cite with an update once in the text. For example, the cited Table 2 (line 93) should be cited by Table 1, and Table 1 (line 401, 393) should be changed to Table 5. Please carefully double-check in whole manuscript.

Response: Thank you for your valuable suggestion. The author has relabeled/renumbered all tables and cited with an update in the text. For example, the table 2 (line 123) has been changed to table 1. Table 1 has been changed to Table 5.

Comment: Some Figs (Fig 2, 8) presented in the manuscript were of low quality. The authors should replace it with a new one of high quality. Tip, the Fig data might be placed in the text of the manuscript in high resolution by choosing the paste option as insert the contents of clipboard as files, do not paste as pictures.

Response: Thank you for your valuable suggestion. Figure 2. Figure 8 has been replaced with a new high-quality image and inserted as a file

Comment: Represent the manuscript according to the journal’s style: Section “Abstract” should be removed sentence/paragraph section head (Background, Methods, Results, and Conclusions) since it was presented following other journals’ style (Biomedcentral, BMC). Numbered the subsections and Capitalize Each Word in the paper title and subsection. For example, subsection “Sequence characteristics of the strawberry DFR gene family” (line 90) -> “2.1. Sequence Characteristics of the Strawberry DFR Gene Family”.

Response: Thank you for your valuable suggestion. The manuscript has been presented according to the journal’s style: Sentence/paragraph section head (Background, Methods, Results, and Conclusions) in the Abstract section have been deleted; sections have been numbered in the title and section of the paper has been capitalized. For example, subsection “Sequence characteristics of the strawberry DFR gene family” -> “2.1. Sequence Characteristics of the Strawberry DFR Gene Family”.

Comment: Italicize gene name: DFR (line 77, 78, ect)

Response: Thank you for your valuable suggestion. The gene name DFR has been italicized (line 73, 75, ect).

Comment: Italicize the scientific name of plant: Nicotiana gossei, Arabidopsis thaliana (Fig 3’s legend)

Response: Thank you for your valuable suggestion. The scientific name of plant: Nicotiana gossei, Arabidopsis thaliana (Fig 3’s legend) have been italicized.

Comment: Change the unit of (protein molecular weight) from “D” (line 95) to “Da.

Response: Thank you for your valuable suggestion. The unit of (protein molecular weight) has been changed from “D” to “Da” (line 94).

Comment: Lowercase “Alpha helix”, “Random coil”, “Extended strand”, and “Beta turn” -> “alpha helix”, “random coil”, “extended strand”, and “beta turn”

Response: Thank you for your valuable suggestion. “Alpha helix”, “Random coil”, “Extended strand”, and “Beta turn” have been changed to lowercase.

Comment: Grammar error: “Arabidopsis Thaliana” (line 430) change to “Arabidopsis thaliana”

.

Response: Thank you for your valuable suggestion. “Arabidopsis Thaliana” has been changed to “Arabidopsis thaliana” (line 361).

Reviewer 3 Report

Comments and Suggestions for Authors

The article entitled “Identification of strawberry DFR gene family and expression analysis in different coloring stages of fruit” by Chen et al., is written well and provide basis for further research on DFR genes in strawberry, but there are some revisions that might be incorporated before formal acceptance in the journal.

General Comments

Add the full form of DFR genes in the title.

The revised title may be like; Genome wide Identification of Dihydroflavonol 4-reductase (DFR) gene family and its expression analysis in different coloring stages of fruit in strawberry

The paper needs minor English editing and grammar correction. Authors are directed to check the punctuations and spellings.

When abbreviations are first introduced in the text, they should be written in full and then followed by the abbreviation.

Italicize the gene names as well as the species names.

Abstract

Remove the sub heading like background, methods, results and conclusion from the abstract.

Add a brief significance of DFR genes in strawberry at start to highlight why it is important to study.

Add a brief methodology for the identification of DFR genes.

The results are presented well but not organize, please refer to the papers previously published in IJMS. e.g. https://doi.org/10.3390/ijms23137335

Introduction

The introduction is poorly written.

1st paragraph do not provide the detailed background and significance about DFR family.

Add the detailed significance and function of DFR genes in plants, and add the research gaps of DFR genes in strawberry.

Add the introduction and economic importance of strawberry in the last / 3rd paragraph. Identify the significance of current work and present your hypothesis/ objectives of the study.

Authors are directed to improve this part with recent citations.

Line 61 to 78: Recheck the citation

Results

Results section are written well. However, throughout the results section needs to grammatically check.

Update the legends of Table 1. The legends should be self-explanatory. Add all the studied properties into the legend.

Legends of Table 3 is not explaining the table presented. Write a detailed legends that what is presented in the table, what is shown via the values?

Figure 3. How you named the clades in phylogenetic tree? What is the different names represent? Moreover the figure is stretched. Replace it with high quality image and do not stretch the figure.

Add a result of interaction of these FDR genes with miRNA. Refer to the paper https://doi.org/10.3390/ijms23137335, previously published in IJMS.

Discussion

This section seems shallow, authors are required to provide justification for each result with recent citation.

Materials and Methods

Some section of this section needs proper citations. Authors are directed to provide the most recent citations for the sub-sections and elaborate the methodology so that the reader could understand the protocols followed. For instance see the previously published papers https://doi.org/10.3390/ijms23137335, https://doi.org/10.3390/genes13101766, https://doi.org/10.1007/s11103-019-00955-2.

All gene names should be italicized.

Comments on the Quality of English Language

Minor English editing is required.

Author Response

Dear Editors and Reviewers,

Thank you for your letter and for the comments concerning our manuscript titled “Genome wide Identification of Dihydroflavonol 4-reductase (DFR) gene family and its expression analysis in different coloring stages of fruit in strawberry”. The comments have helped us improve our manuscript and have been valuable for our future study. We have studied the comments carefully and have applied corrections, which we hope would meet your approval. The main corrections in the paper and the response to the comments are as follows:

General Comments

Comment: Add the full form of DFR genes in the title. The revised title may be like; Genome wide Identification of Dihydroflavonol 4-reductase (DFR) gene family and its expression analysis in different coloring stages of fruit in strawberry.

Response: Thank you for your valuable suggestion. The full form of DFR gene has been added to the title.The title has been changed to Genome wide Identification of Dihydroflavonol 4-reductase (DFR) gene family and its expression analysis in different coloring stages of fruit in strawberry.

Comment: The paper needs minor English editing and grammar correction. Authors are directed to check the punctuations and spellings.

Response: Thank you for your valuable suggestion. All the authors checked and corrected the spelling of the text, and at the same time, a native English teacher was invited to revise the grammar of the text.

Comment: When abbreviations are first introduced in the text, they should be written in full and then followed by the abbreviation.

Response: Thank you for your valuable suggestion. Authors have changed the abbreviation to the full name.

Comment: Italicize the gene names as well as the species names

Response: Thank you for your valuable suggestion. The names of genes and species have been italicized.

Abstract

Comment: Remove the sub heading like background, methods, results and conclusion from the abstract.

Response: Thank you for your valuable suggestion. Subtitles such as background, methods, results and conclusions have been deleted from the abstract.

Comment: Add a brief significance of DFR genes in strawberry at start to highlight why it is important to study.

Response: Thank you for your valuable suggestion. The brief meaning of DFR gene has been added at the beginning, emphasizing the importance of studying it.

Comment: Add a brief methodology for the identification of DFR genes.

Response: Thank you for your valuable suggestion. A brief method for identifying DFR genes has been added

Comment: The results are presented well but not organize, please refer to the papers previously published in IJMS. e.g. https://doi.org/10.3390/ijms23137335

Response: Thank you for your valuable suggestion. Changes have been made to the results section with reference to the paper published in the IJMS (https://doi.org/10.3390/ijms23137335).

Introduction

Comment: The introduction is poorly written. 1st paragraph do not provide the detailed background and significance about DFR family.

Response: Thank you for your valuable suggestion. A detailed background and significance about the DFR family has been added in the first paragraph.

Comment: Add the detailed significance and function of DFR genes in plants, and add the research gaps of DFR genes in strawberry.

Response: Thank you for your valuable suggestion. The detailed significance and function of DFR genes in plants have been increased. In addition, the research gap of DFR gene in strawberry was increased.

Comment: Add the introduction and economic importance of strawberry in the last / 3rd paragraph. Identify the significance of current work and present your hypothesis/ objectives of the study.

Response: Thank you for your valuable suggestion. Already in the last paragraph add the introduction and economic importance of strawberries.It also explains the significance and goals of the current work.

Comment: Authors are directed to improve this part with recent citations.

Response: Thank you for your valuable suggestion. The authors of this section have made use of recent citations.

Comment: Line 61 to 78: Recheck the citation

Response: Thank you for your valuable suggestion. The quotation in the article has been updated.

Results

Comment: Results section are written well. However, throughout the results section needs to grammatically check.

Response: Thank you for your valuable suggestion. We invited native English speakers to revise the grammar of the whole text.

Comment: Update the legends of Table 1. The legends should be self-explanatory. Add all the studied properties into the legend.

Response: Thank you for your valuable suggestion. The legend in Table 1 has been updated and all the properties studied have been added to the legend.

Comment: Legends of Table 3 is not explaining the table presented. Write a detailed legends that what is presented in the table, what is shown via the values?

Response: Thank you for your valuable suggestion. The legend in Table 3 has been changed to a detailed legend. For example, in FvDFR1, 42.86 indicates that alpha helix accounts for 42.86% of the protein secondary structure, 13.07 indicates that extended strand accounts for 13.04% of the protein secondary structure, 7.29 indicates that beta turn accounts for 7.29% of the protein secondary structure, and 36.78 indicates that random coil accounts for 36.78% of the protein secondary structure.

Comment: Figure 3. How you named the clades in phylogenetic tree? What is the different names represent? Moreover the figure is stretched. Replace it with high quality image and do not stretch the figure.

Response: Thank you for your valuable suggestion. The phylogenetic tree is named with A, B, C, D, E and F, and these different names represent different subfamilies. Figure 3 has been replaced with a new high-quality image.

Comment: Add a result of interaction of these FDR genes with miRNA. Refer to the paper https://doi.org/10.3390/ijms23137335, previously published in IJMS.

Response: Thank you for your valuable suggestion. The authors carefully read the academic papers recommended by experts, talked about the interaction between PPR gene and miRNA, screened and identified some candidate key miRNAs, and provided support for subsequent miRNA research. In this study, we focused on the interaction between functional genes and between transcription genes and transcription factors, with the aim of proposing a regulatory model with "transcription regulators-functional genes-anthocyanins" as modules, and miRNAs are not currently in the intended purpose of this study. As suggested by the reviewers, miRNA research is also an important part of the content, and future studies should consider paying attention to the complex formed by miRNA and other regulatory factors involved in the regulation of anthocyanins.

Discussion

Comment: This section seems shallow, authors are required to provide justification for each result with recent citation.

Response: Thank you for your valuable suggestion. The discussion section has been changed and new literature has been cited to support this section.

Materials and Methods

Comment: Some section of this section needs proper citations. Authors are directed to provide the most recent citations for the sub-sections and elaborate the methodology so that the reader could understand the protocols followed. For instance see the previously published papers https://doi.org/10.3390/ijms23137335, https://doi.org/10.3390/genes13101766, https://doi.org/10.1007/s11103-019-00955-2.

Response: Thank you for your valuable suggestion. The author makes appropriate up-to-date citations in this section and elaborates on his methodology, such as citing https://doi.org/10.3390/ijms23137335, https://doi.org/10.3390/genes13101766, https://doi.org/10.1007/s11103-019-00955-2.

Comment: All gene names should be italicized.

Response: Thank you for your valuable suggestion.  All gene names have been italicized.

Round 2

Reviewer 1 Report

Comments and Suggestions for Authors

I have read through the revised manuscript.  The authors made some improvements.  However, the manuscript still reads somewhat poorly.

Here are the comments:

1. The tables have been renumbered.  I asked for clarification of units for what is now Table 2 in the revised manuscript and this has not been provided. If these are units of gene expression, then the authors can just say this.

Line 64 and 282:     These are sentence fragments.  Although it is sometimes okay to start a sentence with "And", neither of these is an acceptable instance. Here, in line 64 the verb "found" lacks a subject and thus what is written is a fragment and not permissible.

"And found that HvDFR was constitutive expressed in all tissues and expressed highly in flower as well as was positively correlated with anthocyanin content"

In Line 282, the sentence has subject and verb, but it does not represent a complete thought.  "And others are in a variety of plants."

Comments on the Quality of English Language

Line 64 and 282:     These are sentence fragments.  Although it is sometimes okay to start a sentence with "And", neither of these is an acceptable instance. Here, in line 64 the verb "found" lacks a subject and thus what is written is a fragment and not permissible.

"And found that HvDFR was constitutive expressed in all tissues and expressed highly in flower as well as was positively correlated with anthocyanin content"

In Line 282, the sentence has subject and verb, but it does not represent a complete thought.  "And others are in a variety of plants."

Author Response

Dear Editors and Reviewers,

Thank you for your letter and for the comments concerning our manuscript titled “Genome wide Identification of Dihydroflavonol 4-reductase (DFR) gene family and its expression analysis in different coloring stages of fruit in strawberry”. The comments have helped us improve our manuscript and have been valuable for our future study. We have studied the comments carefully and have applied corrections, which we hope would meet your approval. The main corrections in the paper and the response to the comments are as follows:

Comment: The tables have been renumbered. I asked for clarification of units for what is now Table 2 in the revised manuscript and this has not been provided. If these are units of gene expression, then the authors can just say this.

Response: Thank you for your valuable suggestion. Upon inspection by the authors, the units in Table 2 are as written in the manuscript.

Comment: Line 64 and 282: These are sentence fragments. Although it is sometimes okay to start a sentence with "And", neither of these is an acceptable instance. Here, in line 64 the verb "found" lacks a subject and thus what is written is a fragment and not permissible.

Response: Thank you for your valuable suggestion. These sentences have been changed.

Comment: "And found that HvDFR was constitutive expressed in all tissues and expressed highly in flower as well as was positively correlated with anthocyanin content"

Response: Thank you for your valuable suggestion. This sentence has been changed to read: Qin et al. found that HvDFR was constitutive expressed in all tissues and expressed highly in flower as well as was positively correlated with anthocyanin content( line 64).

Comment: In Line 282, the sentence has subject and verb, but it does not represent a complete thought.  "And others are in a variety of plants."

Response: Thank you for your valuable suggestion. This sentence has been changed to read: The DFR gene family may also be present in other plants( line 282).